# HESI-MS/MS Analysis of Phenolic Compounds from *Calendula aegyptiaca* Fruits Extracts and Evaluation of Their Antioxidant Activities

**DOI:** 10.3390/molecules27072314

**Published:** 2022-04-02

**Authors:** Wafa Grati, Sonda Samet, Bouthaina Bouzayani, Amani Ayachi, Michel Treilhou, Nathan Téné, Raoudha Mezghani-Jarraya

**Affiliations:** 1Laboratory of Organic Chemistry LR17ES08, Natural Substances Team, Faculty of Sciences of Sfax, University of Sfax, P.O. Box 1171, Sfax 3000, Tunisia; wafagrati0609@gmail.com (W.G.); samet.sonda95@gmail.com (S.S.); bouzayanibouthaina@yahoo.com (B.B.); amaniayachi21@gmail.com (A.A.); raoudhajarraya@yahoo.fr (R.M.-J.); 2Equipe BTSB-EA 7417, Institut National Universitaire Jean-François Champollion, Université de Toulouse, Place de Verdun, 81012 Albi, France; michel.treilhou@univ-jfc.fr

**Keywords:** *Calendula aegyptiaca*, LC–MS/MS analysis, phenolic compounds, flavonoids, saponins, antioxidant activity

## Abstract

Considering medicinal plants as an inexhaustible source of active ingredients that may be easily isolated using simple and inexpensive techniques, phytotherapy is becoming increasingly popular. Various experimental approaches and analytical methods have been used to demonstrate that the genus Calendula (Asteraceae) has a particular richness in active ingredients, especially phenolic compounds, which justifies the growing interest in scientific studies on this genus’ species. From a chemical and biological viewpoint, *Calendula aegyptiaca* is a little-studied plant. For the first time, high-performance liquid chromatography combined with negative electrospray ionization mass spectrometry (HPLC-HESI-MS) was used to analyze methanolic extracts of *Calendula aegyptiaca* (*C. aegyptiaca*) fruits. Thirty-five molecules were identified. Flavonoids (47.87%), phenolic acids (5.18%), and saponins (6.47%) formed the majority of these chemicals. Rutin, caffeic acid hexoside, and Soyasaponin βg’ were the most abundant molecules in the fruit methanolic extract, accounting for 17.49% of total flavonoids, 2.32 % of total phenolic acids, and 0.95% of total saponins, respectively. The antioxidant activity of the fruit extracts of *C. aegyptiaca* was investigated using FRAP, TAC, and DPPH as well as flavonoids and total phenols content. Because the phenolic components were more extractable using polar solvents, the antioxidant activity of the methanolic extract was found to be higher than that of the dichloromethane and hexane extracts. The IC50 value for DPPH of methanolic extract was found to be 0.041 mg·mL^−1^. Our findings showed that *C. aegyptiaca* is an important source of physiologically active compounds.

## 1. Introduction

Medicinal plants have a variety of biological and therapeutic properties that are helpful to one’s health and effective in the treatment of a variety of disorders [1]. In both a therapeutic and preventive context, these natural resources have the potential to be a viable alternative to synthetic medications. Tunisia has the most diverse flora in North Africa. This wealth is due to the geographic and climatic changes observed from north to south of the country. Nonetheless, from a phytochemical and biological point of view, this floristic fortune has only briefly been examined. The Asteraceae are angiosperms’ most important family, with around 25,000 species grouped into 1600 genus [2]. *Calendula* is the most well known of the Asteraceae family, with roughly 25 species (*C. officinalis* and *C. arvensis…*). From an economic and medicinal standpoint, this genus is extremely valuable. Indeed, several species of *Calendula* are used in a variety of goods nowadays. It has been shown in the literature to have anti-inflammatory, healing, anticancer, antidiabetic, and gastro-protective activities [3]. Considering the widespread use of this genus, the current research intended to investigate a novel plant, *C. aegyptiaca*. This plant is mainly found in Africa (Tunisia, Egypt, Algeria, Morocco, and Chad) and Europe (Spain and Portugal). Generally, polar extraction solvents such as methanol are used to obtain extracts rich in bioactive metabolites. As a result, the methanolic extract of the fruits of *C. aegyptiaca* was chosen in this work for a structural examination of particular molecules, which are primarily responsible for the antioxidant property described in this paper. Compound identification was performed using HPLC-HESI-MS.

## 2. Results and Discussions

### 2.1. Extraction

The maceration method is based on the degree of solubility of organic molecules in organic solvents [4]. The selection of an appropriate extraction solvent is required for plant materials that include many components. The extract had a dark brownish color in methanol (MeOH), a greenish color in dichloromethane (DCM), and a yellowish color in *n*-hexane. The findings revealed that fruits’ ingredients were best soluble in polar solvents (MeOH). The resulting extract exhibited the highest yield, which was 9.67%. In the *n*-hexane and DCM extracts, lower yields were recorded (Table 1).

### 2.2. Total Phenolics and Flavonoids Contents of Various Extracts from C. aegyptiaca Fruits

Two families of chemicals were detected in all extracts: phenolic acids and flavonoids. Total phenolics and flavonoids contents of *C. aegyptiaca* fruits extracts, expressed in mg of gallic acid equivalent per g of dried extract (mg GAE/g DE) and mg of quercetin equivalent per g of dried extract (mg QE/g DE), respectively, are summarized in Table 2. The MeOH extract had a higher concentration of phenolic acids (275.38 mg GAE/g DE) and flavonoids (204.57 mg QE/g DE) than the DCM and hexane extracts (*p* 0.05).

There was no information on the total phenol and flavonoid contents of *Calendula* species fruits in previous studies. In comparison to *C. arvensis* flowers, total phenols and flavonoids contents of dried MeOH extract did not exceed 118.18 mg GAE/g and 74.14 mg QE/g, respectively [5]. This result indicates the richness of *C. aegyptiaca* fruits in phenols, mainly flavonoids.

### 2.3. Phytochemical Constituents

LC–MS/MS was used to describe and characterize the major metabolites found in *C. aegyptiaca*’s methanolic extract of fruits. Figure 1 depicts the total ion mass chromatogram profile of this extract. Table 3 shows the MS/MS data of the substances which were tentatively identified.

Thirty-five compounds were identified according to the literature data and LC/HESI-MS fragmentation. Peaks **1**–**4**, **14**, and **16** were identified as phenolic acids. Peaks **6**–**12** and **17**–**19** were identified as flavonoids. Peaks **21**, **22**, **25**, **27**, **28**, **34**–**37**, **39**, and **42**–**45** were assigned as saponins. Peaks **24**, **41**, **46**, and **50** were identified as fatty acids.

Peak **1** (T_R_ = 6.31 min, 2.32%) exhibited a molecular ion [M-H]^−^ at *m*/*z* 341, and its MS^2^ fragmentation gave a base peak at *m*/*z* 179 due to the loss of an hexoside moiety [M-H-162]^−^. The obtained fragment was characteristic of deprotonated caffeic acid. Fragments at *m*/*z* 161 and *m*/*z* 135 were characteristic of the loss of water [M-H-162-18]^−^ and carbon dioxide [M-H-162-44]^−^, respectively. Therefore, compound **1** could be attributed to caffeic acid hexoside [6].

Peak **2** (T_R_ = 8.15 min) presented a pseudo molecular ion [M-H]^−^ at *m*/*z* 315. Analysis of MS^2^ spectra of this compound showed fragments at *m*/*z* 153 (base peak) and *m*/*z* 109 corresponding to the loss of an hexose [M-H-162]^−^ and carbon dioxide [M-H-162-44]^−^, respectively. Thus, this compound could be tentatively proposed as Protocatechuic acid-4-*O*-hexoside [7].

Peaks **3**, **4**, and **14** at T_R_ = 8.51, 11.06, and 18.74 min, respectively, resulted in the observation of a common ion at *m*/*z* 191 that could be attributed to quinic acid. Peak **3** was identified as quinic acid due to its molecular ion [M-H]^−^ at *m*/*z* 191 and other characteristic fragments *m*/*z* 173 [M-H-18]^−^ (loss of H_2_O), *m*/*z* 171 [M-20]^−^ (losses of H_2_O and H_2_), *m*/*z* 127 [M-H-64]^−^ (losses of H_2_O, CO_2_ and H_2_), *m*/*z* 109 [M-H-82]^−^ (losses of 2H_2_O, CO_2_ and H_2_), and *m*/*z* 93 [phenol moiety]^−^ [8]. Peak **4** was proved to be chlorogenic acid with a molecular ion [M-H]^−^ at *m*/*z* 353 and MS^2^ fragment ion at *m*/*z*, 191 [9,10]. Peak **14** showed [M-H]^−^ ions at *m*/*z* 515 and produced daughter ions at *m*/*z* 353 (glucose loss), 335 and 317 (caffeoyl quinic acid loss), 299 (water loss), 255 (carbon dioxide loss), and 191 (deprotonated caffeic acid). According to the fragmentation scheme suggested by Michael et al., this compound was identified as 1, 4-di-*O*-caffeoylquinic acid [11].

Peaks **6** and **18**, revealed at T_R_ = 13.42 and 20.88 min, respectively, corresponded to apigenin derivatives. The first, identified as apigenin C-hexoside-C-pentoside [12], generated a parent ion peak [M-H]^−^ at *m*/*z* 563 and daughter ion peaks at *m*/*z* 503 [M-H-C_2_H_4_O_2_]^−^, *m*/*z* 473 [M-H-C_3_H_6_O_3_]^−^, *m*/*z* 443 [M-H-C_4_H_8_O_4_]^−^, *m*/*z* 383 [M-H-C_3_H_6_O_3_-C_2_H_4_O_2_-CH_2_O]^−^, and *m*/*z* 353 [M-H-C_4_H_8_O_4_-C_3_H_6_O_3_]^−^. The second, with the same molecular ion, was identified as apigenin-*O*-hexosylpentosyl [13], and its fragmentation led to a fragment ion at *m*/*z* 401 [M-H-162]^−^ resulting from hexose loss.

Peaks **7**–**9**, **11**, and **19** were identified as quercetin derivatives due to a characteristic fragment ion at *m*/*z* 301. Peak **7** (T_R_ = 13.82, 8.8%) showed [M-H]^−^ at *m*/*z* 625. This deprotonated molecular ion generated [M-H-C_12_H_20_O_10_]^−^, [M-H-C_12_H_20_O_10_-CH_2_O]^−^, and [M-H-C_12_H_20_O_10_-CO-H_2_O]^−^ ions at *m*/*z* 301, 271, and 255, respectively. This compound was tentatively identified as Quercetin-3,4′-di-*O*-glucoside [14]. Peak **8** (T_R_ = 14.96, 17.49%) generated its [M-H]^−^ ion at *m*/*z* 609. A series of fragment ions appeared at *m*/*z* 343, 301, 300, 271, and 255. According to the literature data, this compound could be identified as rutin [10,14]. Peak **9** (T_R_ = 15.65, 7.57%) produced its [M-H]^−^ ion at *m*/*z* 463. Further fragmentation produced the [M-H-162]^−^ ion at *m*/*z* 301, which correspond to the loss of glucose. Thus, compound **9** was deduced as Quercetin-3-*O*-glucoside [15,16]. Peak **11** (T_R_ = 16.96) could be attributed to Quercetin-*O*-acetyl-glucoside. Indeed, a molecular ion was observed in this molecule at *m*/*z* 505. Further fragmentation led to [M-H-42]^−^ and [M-H-162]^−^, corresponding to characteristic fragmentations of glucose [17]. Peak **19** (T_R_ = 21.11) gave a pseudo molecular ion at *m*/*z* 301 [M-H]^−^. MS^2^ ions were observed at *m*/*z* 179 [M-H-122]^−^ (resulted from a Retro-Diels–Alder cleavage fragmentation) and *m*/*z* 151 [M-H-150]^−^. Then, compound **19** was identified as quercetin [8]. The presence of these chemicals at high levels could be responsible for the antioxidant activity verified in this paper. 

Three peaks, **10**, **12**, and **17**, with main MS^2^ fragmentation ions at *m*/*z* 315 were attributed to isorhamnetin derivatives. Peak **10** (T_R_ = 16.42, 6.15%) was identified as isorhamnetin-3-*O*-rutinoside with [M-H]^−^ ion at *m*/*z* 623 and MS^2^ fragmentation at *m*/*z* 315 [M-H-308]^−^ due to the loss of rutinose, *m*/*z* 300 and 271 characteristic of isorhamnetin aglycone fragmentation [18]. Peak **12** (T_R_ = 17.13) had [M-H]^−^ at *m*/*z* 477. The molecular ion fragmentation yielded ions at *m*/*z* 357, 315, and 314, corresponding to characteristic fragmentations of the glucose moiety. Thus, compound **12** was tentatively identified as isorhamnetin 3-*O*-glucoside [19]. Peak **17** (T_R_ = 19.93) exhibited a molecular anion at *m*/*z* 491 and MS^2^ fragments at *m*/*z* 459, 447, 323, and 315. The last corresponded to the loss of 176 mass unit, which is characteristic, according to the literature data, to glucuronide moiety. Therefore, this compound could be assigned as isorhamnetin-3-*O*-glucuronide [20,21].

Peak **16** (T_R_ = 19.38 min) produced an [M-H]^−^ ion at *m*/*z* 137. In the MS^2^ spectrum, the predominant ion was revealed at *m*/*z* 93 [M-H-44]^−^, which gave the proof for a carbon dioxide unit loss. Then, compound **16** was identified as p-hydroxybenzoic acid [22].

Peaks **21**, **22**, **32**, **35**, and **36** presented a common fragment ion at *m*/*z* 471 characteristic of hedragenin derivatives. Peak **21** (T_R_ = 22.42 min) corresponded to betavulgaroside VI with a pseudo molecular ion [M-H]^−^ at *m*/*z* 971 and MS^2^ fragments at *m*/*z* 851 (loss of C_4_H_8_O_4_), 809 (loss of hexose), and 629 (loss of hexose and hydrated hexose) [23]. Peak **22** (T_R_ = 22.74 min) presented a molecular ion [M–H]− at *m*/*z* 809. The fragmentation of this saponin yielded daughter ions at *m*/*z* 689, 647, 629, and 471. These ions corresponded to [M-H-C_4_H_8_O_4_]^−^, [M-H-hexose]^−^, [M-H-hydrated hexose]^−^, and [aglycon-H]^−^, respectively, as illustrated in Figure 2a. Thus, compound **22** was identified as gluco-glucuronic acid hedragenin [24]. Peak **32** (T_R_ = 28.01 min), with its molecular ion [M-H]- at *m*/*z* 647 and MS^2^ fragments at 629 (water loss), 571 (water and C_2_H_2_O_2_ losses), and 471 (deprotonated hedragenin), was identified, according to the literature, as glucuronic acid hedragenin [25]. Peak **35** (T_R_ = 28.95min) exhibited a parent ion [M-H]^−^ at *m*/*z* 777 with daughter ions at *m*/*z* 633 [M-H-dehydrated hexose]^−^, *m*/*z* 615 [M-H-hexose]^−^, and *m*/*z* 471 [aglycon-H]^−^. This compound was then identified as hedragenin dihexoside. Peak **36** (T_R_ = 29.21 min), which presented [M-H]^−^ at *m*/*z* 791, presented the same profile of fragmentation of saponin **22**. Thus, this compound could be assigned as dehydrated gluco-glucuronic acid hedragenin.

Four peaks, **23**, **41**, **46,** and **50,** were attributed to fatty acids. Peaks **23**, **41**, and **50,** revealed at T_R_ = 23.35, 30.81, and 34.17 min, respectively, were identified as octadecenoic acid derivatives. These compounds showed a common MS^2^ daughter ion at *m*/*z* 171 corresponding to the fragment OOC(CH_2_)_7_-CH-OH. The literature data proved that compound **23** ([M-H]^−^ at *m*/*z* 327) presented a mixture of 9-*oxo*-12,13-dihydroxy-10-octadecenoic and 13-*oxo*-9,10-dihydroxy-11-octadecenoic acids [26,27]. For compound **41** ([M-H]^−^ at *m*/*z* 313), MS^2^ spectra showed consecutive losses of water molecules as well as aliphatic residues. This compound was assigned to be dihydroxyoctadecenoic acid [19]. Peak **50**, exhibiting an [M-H]^−^ ion at *m*/*z* 279 and MS^2^ fragments at *m*/*z* 261 (water loss), *m*/*z* 235 (carbon dioxide loss), and *m*/*z* 171, could be tentatively identified as deprotonated linoleic acid [28]. Peak **46** observed at T_R_ = 33.05 min presented a pseudo molecular ion [M-H]^−^ at *m*/*z* 295. MS^2^ fragmentation showed characteristic peaks at *m*/*z* 277 (water loss), *m*/*z* 251 (carbon dioxide loss), *m*/*z* 171. Then, compound **46** was identified according to Seon et al. as 9-hydroxy-10,12-actadecadienoic acid [29]. 

Eight peaks, **25**, **27**, **29**, **34**, **42**, **43**, **44**, and **45**, were identified as oleanolic acid saponins derivatives. MS^2^ analysis of those saponins showed a typical fragment ion at *m*/*z* 455 corresponding to deprotonated oleanolic acid aglycone. Peak **25** (T_R_ = 24.41 min) presented a base peak at *m*/*z* 955 and exhibited *m*/*z* 793 [M-H-162]^−^ (loss of glucose), *m*/*z* 613 [M-H-342]^−^ (loss of water and two glucose moieties), and *m*/*z* 455 ions in the MS^2^ spectra. By comparing it with the literature data, this compound was tentatively identified as ginsenoside Ro [30]. Saponin **27** (T_R_ = 25.02 min) afforded a pseudo molecular ion [M-H]^−^ at *m*/*z* 925. Based on the MS^2^ fragmentation, this molecule consisted of the oleanolic acid aglycone ion at *m*/*z* 455 with sugar unit structures (uronic acid, hexose, and pentose) which were linked in different positions, as demonstrated in Figure 2b [23,31]. Saponins **29** (T_R_ = 26.15 min) and **34** (T_R_ = 28.69 min) showed [M-H]^−^ ions at *m*/*z* 793 and 835, respectively. Both compounds presented the same profile of MS^2^ fragmentation *m*/*z* 673, 631, 613, and 569. For saponin **29**, these fragment ions corresponded to consecutive losses of 120 amu (characteristic fragmentation of hexose), 162 amu (hexose loss), 180 amu (hydrated hexose loss), and 224 amu (carbon dioxide with hydrated hexose losses). Compound **29** was then identified according to Faustino et al. as calenduloside G [13], while compound **34** could be tentatively identified as calenduloside G derivative. Peak **42** (T_R_ = 31.30 min) and **44** (T_R_ = 32.02 min) presented molecular ions [M-H]^−^ at *m*/*z* 617 and 761, respectively. MS^2^ fragmentation of compound **42** showed fragment ions at *m*/*z* 599 (water loss), *m*/*z* 571 (water and carbone dioxide losses), *m*/*z* 497 (-120 amu characteristic of hexose fragmentation), and *m*/*z* 455, corresponding to deprotonated aglycone oleanolic acid. This saponin could therefore be attributed to oleanolic acid 28-*O*-β-D-glucopyranoside isolated by Kumar et al. [32]. MS^2^ fragmentation of compound **44** showed the same fragment ions as compound **42** and could be assigned as oleanolic acid dihexoside. To the best of our knowledge, the last two compounds were fragmented using LC-HESI-MS for the first time in the current paper. Compound **43** (T_R_ = 31.62 min) was identified by Mroczek et al. as 3-*O*-β-D glucuruopyranosyl of oleanolic acid due to its [M-H]^−^ ion at *m*/*z* 631 and fragment ion at *m*/*z* 455 (oleanolic acid aglycone) obtained after the loss of 176 amu (glucuronic acid loss) [24]. Compound **45** (T_R_ = 32.27 min), [M-H]^−^ at *m*/*z* 775, presented the same structure of compound **43** with a supplement hexose moiety, as evidenced by its MS^2^ fragmentation.

Peaks **37** and **39** were proposed to be soyasapogenol E derivatives with characteristic fragment ions attributed to the aglycone at *m*/*z* 455. Saponin **37** (T_R_ = 29.5 min) showed the same molecular ion as saponin **30**, [M-H]^−^ ion at *m*/*z* 793 and exhibited MS^2^ fragment ions at *m*/*z* 775 (water loss), *m*/*z* 731 (water and carbone dioxide losses), and *m*/*z* 613 (hexose and water losses). Therefore, this compound was assigned according to Nascimento et al. as soyasaponin βe’ [33]. In addition, soyasaponin βg’ (compound **39**, T_R_ = 30.34 min) was also identified in the same paper with a parent ion [M-H]^−^ ion at *m*/*z* 763 and characteristic MS^2^ fragment ions at *m*/*z* 719, 701, 613, 523, and 455.

**Table 3 molecules-27-02314-t003:** Compounds from *C. aegyptiaca* fruits (MeOH extract) identified through LC–MS/MS (negative mode).

Compound	T_R_ (min)	Relative Abundance (%)	[M-H]^−^ (*m*/*z*)	Molecular Formula	LC/HESI-MS^2^ (*m*/*z*)	Tentative Identification	Reference
**1**	6.31	2.32	341	C_15_H_18_O_9_	179 (100), 161, 135	Caffeic acid hexoside	[6]
**2**	8.15	1.94	315	C_13_H_16_O_9_	153 (100), 109	Protocatechuic acid-4-*O*-hexoside	[7]
**3**	8.51	0.04	191	C_7_H_12_O_6_	173, 171, 127 (100), 109, 93	Quinic acid	[8]
**4**	11.06	0.73	353	C_16_H_18_O_9_	191 (100)	Chlorogenic acid	[9,10]
**5**	13.01	1.29	507	-	325 (100), 181	Unidentified	-
**6**	13.42	1.49	563	C_26_H_28_O_14_	503, 473, 443 (100), 383, 353	Apigenin *C*-hexoside- *C*-pentoside	[12]
**7**	13.82	8.80	625	C_27_H_30_O_17_	301 (100), 271, 255	Quercetin-3,4′-di-*O*-glucoside	[14]
**8**	14.96	17.49	609	C_27_H_30_O_16_	343, 301 (100), 300, 271, 255	Rutin	[10,14]
**9**	15.65	7.57	463	C_21_H_20_O_12_	301 (100)	Quercetin-3-*O*-glucoside	[15,16]
**10**	16.42	6.15	623	C_28_H_32_O_16_	315 (100), 300, 271	Isorhamnetin-3-*O*-rutinoside	[18]
**11**	16.96	5.13	505	C_23_H_22_O_13_	463, 301 (100)	Quercetin-*O*-acetyl glucoside	[17]
**12**	17.13	0.88	477	C_22_H_22_O_12_	357, 315, 314 (100)	Isorhamnetin 3-*O*-glucoside	[19]
**13**	18.38	1.27	461	-	323 (100), 137	Unidentified	-
**14**	18.74	0.08	515	C_25_H_24_O_12_	353 (100), 335, 317, 299, 255,191, 173	1,4-di-*O*-caffeoylquinic acid	[11]
**15**	18.81	0.14	465	-	297 (100), 183	Unidentified	-
**16**	19.38	0.07	137	C_7_H_6_O_3_	93 (100)	p-Hydroxybenzoic acid	[22]
**17**	19.93	0.17	491	C_22_H_20_O_13_	459, 447, 323, 315 (100)	Isorhamnetin-3-*O*-glucuronide	[21,22]
**18**	20.88	0.14	563	C_26_H_28_O_14_	401 (100)	Apigenin-*O*-hexosylpentosyl	[13]
**19**	21.11	0.05	301	C_15_H_10_O_7_	179 (100), 151	Quercetin	[8]
**20**	21.44	3.35	1165	-	1146, 1002 (100), 657, 463	Unidentified	-
**21**	22.42	0.14	971	C_47_H_72_O_21_	851, 809 (100), 629	Betavulgaroside VI	[23]
**22**	22.74	0.20	809	C_42_H_66_O_15_	791, 689, 647 (100), 629,471	Gluco-glucuronic acid hedragenin	[24]
**23**	23.35	2.00	327	C_18_H_32_O_5_	291, 229 (100), 211, 209, 171	*Oxo*-dihydroxy-octadecenoic acid	[26,27]
**24**	23.71	13.02	1149	-	1131,1048,970 (100), 839,444	Unidentified	-
**25**	24.41	0.34	955	C_48_H_76_O_19_	793 (100), 613, 455	Ginsenoside Ro	[30]
**26**	24.66	2.9	987	-	925,825 (100), 791, 543	Unidentified	-
**27**	25.02	0.19	925	C_47_H_73_O_18_	805, 763 (100), 613	Hexose-pentose uronic acid oleanolic acid	[23,31]
**28**	25.85	5.26	1027	-	1009, 983 (100), 966	Unidentified	-
**29**	26.15	0.78	793	C_42_H_66_O_14_	673, 631 (100), 613, 569, 455	Calenduloside G	[13]
**30**	26.80	0.25	695	-	533 (10), 371	Unidentified	-
**31**	27.46	5.33	937	-	793 (100)	Unidentified	-
**32**	28.01	0.94	647	C_36_H_56_O_10_	629, 571, 471 (100)	Glucuronic acid hedragenin	[25]
**33**	28.30	0.69	987	-	969, 841, 824 (100), 816, 614	Unidentified	-
**34**	28.69	0.25	835	-	793, 775, 673, 613, 569 (100), 455	Calenduloside G derivative	-
**35**	28.95	0.45	777	C_42_H_68_O_14_	633 (100), 615, 471	Hedragenin dihexoside	-
**36**	29.21	0.34	791	C_42_H_64_O_14_	689, 647 (100), 629	Dehydrated gluco-glucuronic acid Hedragenin	-
**37**	29.5	0.12	793	C_42_H_65_O_14_	775, 731, 613 (100), 455	Soyasaponin βe’	[33]
**38**	30.01	0.39	793	-	613, 551, 483 (100), 455	Unidentified	-
**39**	30.34	0.95	763	C_41_H_63_O_13_	719, 701, 613 (100), 523, 455	Soyasaponin βg’	[33]
**40**	30.60	1.74	675	-	415, 937 (100), 305, 235	Unidentified	-
**41**	30.81	0.41	313	C_18_H_34_O_4_	295, 277, 201 (100), 171	Dihydroxyoctadecenoic acid	[19]
**42**	31.30	0.17	617	C_36_H_58_O_8_	599, 571, 497, 455 (100)	Oleanolic acid 28-*O*-β-D-glucopyranoside	[32]
**43**	31.62	0.52	631	C_36_H_56_O_9_	613, 455(100)	3-*O*-β-D glucuruopyranosyl of oleanolic acid	[24]
**44**	32.02	0.28	761	C_42_H_66_O_12_	617, 599 (100), 571, 497, 455	Oleanolic acid dihexoside	-
**45**	32.27	0.80	775	C_42_H_64_O_13_	631 (100), 613, 455	Gluco- glucuruopyranosyl of oleanolic acid	-
**46**	33.05	0.03	295	C_18_H_32_O_3_	277 (100), 251, 171	9-Hydroxy-10,12-actadecadienoic acid	[29]
**47**	33.12	0.03	527	-	509, 277 (100), 249	Unidentified	-
**48**	33.38	0.12	564	-	504 (100)	Unidentified	-
**49**	33.50	0.09	504	-	279 (100)	Unidentified	-
**50**	34.17	0.01	279	C_18_H_32_O_2_	261 (100), 235, 171	Linoleic acid	[28]

### 2.4. Antioxidant Activity of Fruit Extracts In Vitro

The total antioxidant capacity (TAC) of the extracts (Table 4) was calculated using the phosphomolybdenum method. By forming a green phosphomolybdenum complex (V) with a maximum absorbance at 695 nm, the antioxidant compounds converted Mo(VI) to Mo(V). MeOH extract had the highest antioxidant capacity (253.394 mg gallic acid equivalents (GAE/g extract), followed by DCM (181.414 mg GAE/g extract) and *n*-hexane (123.771 mg GAE/g extract) extracts, which could be explained by its high levels of total phenolic acids and flavonoids contents (Table 2).

The DPPH scavenging activity of phenols and flavonoids was also investigated (Table 4). When compared to DCM and *n*-hexane extracts (IC_50_ = 0.050 mg·mL^−1^ and IC_50_ = 0.054 mg·mL^−1^), which presented a moderate and low significance, respectively, the MeOH extract of *C. aegyptiaca* had significantly higher DPPH scavenging activity (IC_50_ = 0.041 mg·mL^−1^). Our findings suggest that hydroxyl groups could intervene as electron donors, transforming free radicals into much more stable substances by scavenging radicals. According to the literature [34], the methanolic extract of this plant had higher DPPH scavenging activity (IC_50_ = 0.041 mg·mL^−1^) than the hydro-methanol extract of *C. officinalis* leaves (0.57 mg·mL^−1^) and lower than that of flowers (0.35 mg·mL^−1^).

The Ferric reducing activity power (FRAP) method is based on electron-donating antioxidants reducing the Fe^3+^ tripyridyltriazine complex (colorless complex) to Fe^2+^-tripyridyltriazine (blue complex) at low pH. The reducing power of extracts and vitamin C was determined (Figure 3). The FRAP test revealed an increase in absorbance with increasing doses of the tested extracts, which corresponded to an increase in reducing power. The obtained results revealed that the extracts’ reducing power increased in direct proportion to their concentration. Because of its highest levels of phenolic and flavonoid content (Table 2), MeOH extract had the highest reducing power (*p* < 0.05), followed by DCM and *n*-hexane extracts.

### 2.5. Correlations

To evaluate the influence of phytochemical constituents on antioxidant capacity, the correlations between the phenolics and flavonoids contents and antioxidant activity of extracts were measured. Table 5 shows different correlations between all extracts; nonetheless, we found strong linear correlations with the respective coefficient of R^2^ = 0.994 (FRAP-TPC), R^2^ = 0.905 (TFC-TPC), R^2^ = 0.941 (TAC-TFC), R^2^ = 0.921 (TAC-DPPH), and R^2^ = 0.969 (TAC-FRAP), and moderate linear correlations with the respective coefficient of R^2^ = 0.859 (TFC-DPPH), R^2^ = 0.884 (TPC-DPPH), R^2^ = 0.861 (TFC-FRAP), and R^2^ = 0.866 (FRAP-DPPH).

## 3. Materials and Methods

### 3.1. Plant Material

*C. aegyptiaca* fruits were collected from Sfax south Tunisia in March 2020, placed in the shade in a well-ventilated area with low humidity (22–25%) at a temperature range of 18–25 °C for 21 days, and then crushed. The plant was recognized by Pr. Mohamed Chaieb [35], Biology Department Faculty of Sciences of Sfax, and a voucher specimen (LCSN150) was stored at the herbarium of the Laboratory of Organic Chemistry (LR17-ES08), Faculty of Sciences, University of Sfax, Tunisia.

### 3.2. Extraction

The dried fruits were crushed in a grinder from Fritsch Company (reference 14.3000.00) in order to obtain much finer particles (2, 3 mm) and then stored in airtight jars away from humidity at room temperature. The moisture content of fruits was evaluated to be 19.71%. The obtained powder was extracted successively with organic solvents of increasing polarities (*n*-hexane, dichloromethane and methanol) with mechanical stirring (plant material/solvent ratio 1:8 (*w*/*v*)). Each extraction was carried out three times at room temperature and for 24 h each time. The macerates were then filtered and evaporated under vacuum to concentrate the extracts. The evaporation process resulted in crude extracts that had no moisture content.

### 3.3. Determination of Phenolic Content

The spectrophotometric method was used to determine the total phenol content (TPC) [36]. A total of 0.5 mL of Folin–Ciocalteu reagent was added to a solution containing 1 mL of a known concentration extract (1 mg·mL^−1^) and 3 mL of distilled water. After 5 min, 0.5 mL of 2% aqueous sodium carbonate (Na_2_CO_3_) was added. After 90 min of incubation at 25 °C, the absorbance at 760 nm was measured. The test was carried out three times. A standard gallic acid graph was used to calculate TPC, which was expressed in milligrams of gallic acid equivalent per gram of dry weight of extract.

### 3.4. Determination of Flavonoid Content

The method established by Heimler et al. [37] was used to determine total flavonoid content (TFC). The approach is based on the creation of a very stable combination between aluminum chloride and the oxygen atoms found on the flavonoids’ carbons 4 and 5, with a maximum absorbance of 430 nm. The calibration curve was generated using quercetin (commercial, Sigma-Aldrich, St. Louis, MO, USA). An amount of 1 mL of 2% aluminum trichloride (AlCl_3_) was blended with 1 mL of sample (1 mg·mL^−1^). The absorbance of the mixture was measured at 430 nm with a spectrophotometer after 15 min of incubation at room temperature. TFC was measured in milligrams of quercetin equivalent (QE) per gram of extract. The experiment was repeated three times.

### 3.5. Antioxidant Activity 

#### 3.5.1. Free Radical Scavenging Activity

The DPPH test was used to assess the extracts’ capacity to scavenge free radicals, as described earlier [38]. DPPH radicals were absorbed at 517 nm; however, absorbance dropped when they were reduced by an antioxidant agent. The decrease in absorbance at 515 nm was measured using UV spectrometry. For concentrations of 0.063, 0.125, 0.25, 0.5, and 1 mg·mL^−1^ of plant extract, vitamin C was employed as a positive control, and all tests were carried out three times. For the assay, different concentrations were used. A total of 2 mL of the DPPH solution and 2 mL of the sample were mixed and left to react in the dark at 37 °C for 30 min as well as a blank test. The results of radical scavenging tests were expressed as 50% inhibition concentration (IC_50_).

#### 3.5.2. Total Antioxidant Capacity

Total antioxidant capacity of the extracts was assessed using the method of phosphomolybdenum complex formation [39]. The reduction of ammonium molybdate and the transmission of electrons are the basis of this approach. A green ammonium phosphate/molybdate complex formed during the process. In total, 1 mL of the reagent solution (sodium phosphate, sulfuric acid, and ammonium molybdate) was combined with 0.1 mL of the sample. The mixes were then incubated for 1 h 30 min in boiling water (95 °C). After the samples cooled, the absorbance was determined at 695 nm. The total antioxidant capacity was expressed as mg of gallic acid equivalents per g of extract. The test was performed in triplicate.

#### 3.5.3. Reducing Power Assay

The procedure used was that of Barros et al. [40]. At various concentrations, 1 mL of each sample was treated with a mixture of potassium ferricyanide (1%) and sodium phosphate (0.2 M). The mixtures were incubated at 50 °C for 20 min. The trichloroacetic acid was then added, and the mixture was placed in the centrifuge for 10 min. After recovery, the supernatant of each mixture was mixed with the ferric chloride solution 0.1% in 2.5 mL of distilled water. Every test was performed three times.

### 3.6. LC-HESI-MS

Fruit methanolic extract of *C. aegyptiaca* was investigated using a Thermo Scientific LTQ XL Mass Spectrometer fitted with a hot electrospray ionization source in the negative mode. Thermo Xcalibur software was used to record ion spectra. A C_18_ reversed phase Luna column at 30 °C (5 µm, 150 mm × 2.1 mm) was delivered to Vanquish HPLC (Thermo Scientific Inc., Waltham, MA, USA) for analysis. A: 0.1% formic acid in water (5% ACN), *v*/*v* and B: 0.1% formic acid in acetonitrile, *v*/*v*, were the selected solvents. The elution gradient was set from 0 to 40% of B during 40 min, 100% B after 50 min, and the column was re-equilibrated between individual runs. The mobile phase had a flow rate of 0.2 mL·min^−1^, and the injection volume was 20 µL. The ion spray voltage was fixed at 3.5 V, the ESI source and the capillary temperature was calibrated at 300 °C, and the sheath and auxiliary gas pressures were set to 50 and 5 psi, respectively. The spectral range was from *m*/*z* 50 to 1200. The approach combined full scans and MS/MS experiments using a collision energy ranging from 10 to 35 eV, depending on the molecular mass of compounds.

### 3.7. Statistical Analysis

A one-way ANOVA was used to assess statistical significance followed by Tukey’s post hoc test for multiple comparisons with *p* = 0.05 and correlation coefficients (r). The Statistical Product and Service Solutions application (SPSS) version 20 was used to conduct these analyses.

## 4. Conclusions

The HPLC–HESI–MS^n^ method was effectively established in this study for the quick separation and identification of various chemicals in the methanol extract of *C. aegyptiaca* fruits. Thirty-five chemicals were identified: six phenolic acids (compounds **1**–**4**, **14**, and **16**), then flavonoids including apigenin derivatives (compounds **6** and **18**), quercetin derivatives (compounds **7**–**9**, **11**, and **19**) and isorhamnetin derivatives (compounds **10**, **12**, and **17**), four fatty acids (compounds **24**, **41**, **46**, and **50**), and fifteen saponins. Oleanolic acid derivatives and hedragenin derivatives were the most commonly reported saponins. As far as we know, compounds **34**–**36**, **44**, and **45** were described for the first time for this species in this paper. Oleanolic acid saponins are known to have anti-inflammatory, anticancer, antihepatotoxic, antidiabetic, and cytotoxic properties. MeOH extract had the highest total phenolic content, as well as the highest total flavonoid contents (275.38 ± 0.39mg GAE/g DE and 204.57 ± 4.101 mg QE/g DE, respectively). These findings imply that phenolic acids (particularly caffeic acid, which accounts for 2.32%) and flavonoids (rutin 17.57%, quercetin-3,4′-di-*O*-glucoside 8.8%, quercetin-3-*O*-glucoside 7.57%) could be responsible for this plant’s antioxidant properties. As a result, fruits of *C. aegyptiaca* should be thought of as a novel source of bioactive compounds with potential applications in a variety of fields. However, more research is required to investigate additional biological activities.

## Figures and Tables

**Figure 1 molecules-27-02314-f001:**
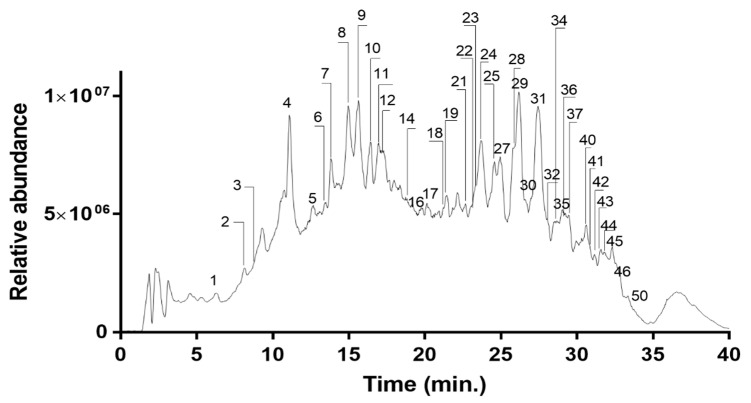
Total ion chromatogram of *C. aegyptiaca* methanolic extract in negative mode.

**Figure 2 molecules-27-02314-f002:**
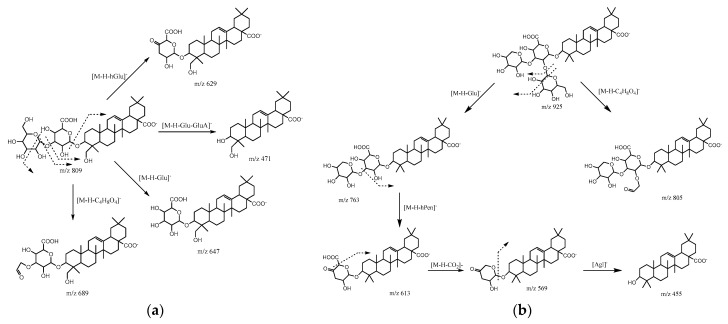
Fragmentation pathway of saponin **22** [M-H]^−^ at *m*/*z* 809 (**a**) and saponin **27** [M-H]^−^ at *m*/*z* 925 (**b**). Glu: glucose, hGlu: hydrated glucose, GluA: glucuronic acid, hPent: hydrated pentose, Agl: aglycone.

**Figure 3 molecules-27-02314-f003:**
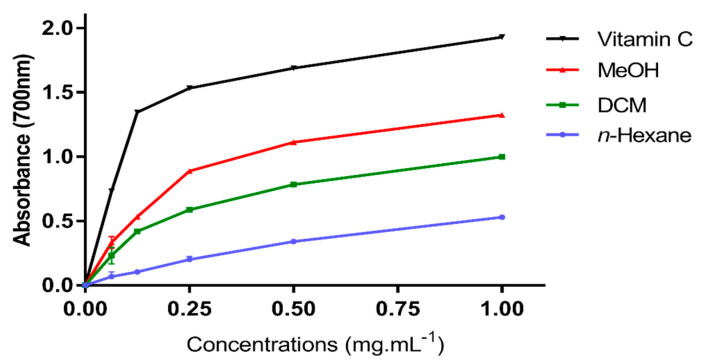
Ferric reducing antioxidant power (FRAP) assays (*n* = 3) compared to vitamin C as standard; the differences were analyzed using Duncan and Tukey’s post hoc test for multiple comparisons with *p* < 0.05.

**Table 1 molecules-27-02314-t001:** Yields (%) of *C. aegyptiaca* fruit extracts.

Extracts	Yields (%)
***n*-Hexane**	0.93
**DCM**	0.47
**MeOH**	9.67

**DCM:** dichloromethane; **MeOH:** methanol.

**Table 2 molecules-27-02314-t002:** Total phenolics and flavonoids contents of fruit extracts of *C. aegyptiaca*.

Extracts	TPC (mg GAE/g DE)	TFC (mg QE/g DE)
***n*-Hexane**	93.37 ± 2.10 ^c^	66.46 ± 9.52 ^c^
**DCM**	190.16 ± 3.21 ^b^	105.18 ± 4.69 ^b^
**MeOH**	275.38 ± 0.39 ^a^	204.57 ± 4.10 ^a^

Values expressed are means ± S.D (n = 3). **TPC:** total phenols content; **TFC:** total flavonoids content; **GAE**: gallic acid equivalent; **QE:** quercetin equivalent; **DE**: dried extract. The differences were analyzed using Duncan and Tukey’s post hoc test for multiple comparisons with *p* < 0.05. a: strong significance, b: modest significance, c: low significance.

**Table 4 molecules-27-02314-t004:** Total antioxidant capacity and DPPH scavenging activity of different extracts of *C. aegyptiaca* fruits.

Extracts	TAC (mg GAE/g DE)	DPPH IC50 (mg·mL^−1^)
***n*-Hexane**	123.771 ± 2.011 ^c^	0.054 ± 0.010 ^c^
**DCM**	181.414 ± 3.044 ^b^	0.050 ± 0.002 ^b^
**MeOH**	253.394 ± 1.198 ^a^	0.041 ± 0.001 ^a^
**Vit C**	-	0.033 ± 0.001 ^a^

Values expressed are means ± S.D (n = 3). **TAC:** total antioxidant capacity, **GAE:** gallic acid equivalent, **DE**: dried extract. **IC_50_** (mg·mL^−1^): inhibition concentration at which 50% of the DPPH (2,2-Diphenyl-1-picrylhydrazyl) are inhibited. The differences were analyzed using Duncan and Tukey’s post hoc test for multiple comparisons with *p* < 0.05. a: strong significance, b: high modest significance, c: low significance.

**Table 5 molecules-27-02314-t005:** Pearson’s determination coefficients (R^2^) for the extracts’ examined parameters.

	TPC	TFC	DPPH	FRAP	TAC
**TPC**	1	-	-	-	-
**TFC**	0.905	1	-	-	-
**DPPH**	0.884	0.859	1	-	-
**FRAP**	0.994	0.861	0.866	1	-
**TAC**	0.987	0.941	0.921	0.969	1

Pearson’s determination coefficients using the 95% confidence interval. **TPC**: total phenolics content, **TFC**: total flavonoids content, **DPPH**: DPPH scavenging activity assay, **FRAP**: Ferric reducing antioxidant power assay, **TAC**: total antioxidant capacity. The Pearson correlation coefficients (R^2^) between different parameters (*p* < 0.05) are shown in the statistical data.

## Data Availability

Not applicable.

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
