# Peer review of "HESI-MS/MS Analysis of Phenolic Compounds from Calendula aegyptiaca Fruits Extracts and Evaluation of Their Antioxidant Activities"

_molecules, 2022, doi:10.3390/molecules27072314_

Round 1
Reviewer 1 Report
Manuscript titled “HESI-MS/MS Analysis of Phenolic Compounds from Calendula aegyptiaca Fruits Extracts and Evaluation of their Antioxidant Activities” (molecules-1659762) describes the quantification of phenolic compounds, flavonoids and antioxidant activities of a calendula species found in Tunisia, whose presence in the literature is said to be scarce. Sensitive analytical techniques (HPLC-HESI-MS/MS) were used to identify 35 out of 50 compounds detected, with most of them being phenolic compounds. The authors conclude that the species analyzed could be a novel source of bioactive compounds, but whose bioactivities remain to be studied. The work is relevant, uses adequate methods to reach the stated goals, and provides a clear description of the data generated. There are some recommendations and suggestions to improve the authors’ work before publishing:
- The abstract provides an ample justification for the study, but it contains minimal information about the data actually reported in the manuscript, its analysis or relevance. Thus, please consider providing additional data (numerical data in particular), which makes it clear what is being reported and the knowledge gained from the work. For example, listing some of the most abundant molecules identified or the effect of solvent on antioxidant capacity (or any other finding that the authors deem significant) will serve to make the abstract more informative and representative of the data being reported.
- Table 2 lists the total phenolic and total flavonoid content of the analyzed samples, however, its title mentions a “phytochemical screening”. This term appears to be too broad, since only phenolics and flavonoids (a particular type of phenolic) are reported. Please consider renaming this table to more accurately reflect its contents.
- In Table 4, please homogenize the number of digits between a value and its error, and be consistent for all data presented on this table.
- Line 282 states that the samples were “dried for 21 days and then crushed”. Please provide more detailed information about the drying method used (temperature, equipment used, exposure to light/oxygen, etc.). Same comment for the crushing procedure. This is particularly relevant, since the phenolic composition and concentration of vegetable matter may be significantly altered during processing, and could be an important variable for other authors who may study this plant in the future.
- In line 287, the crushed fruits are said to have been “extracted successively with 4 L of solvents”. Instead of specifying the absolute volume of solvent used, please provide the mass-to-solvent ratio, in order to make it easier for others to reproduce your extraction method.
Author Response
- The abstract provides an ample justification for the study, but it contains minimal information about the data actually reported in the manuscript, its analysis or relevance. Thus, please consider providing additional data (numerical data in particular), which makes it clear what is being reported and the knowledge gained from the work. For example, listing some of the most abundant molecules identified or the effect of solvent on antioxidant capacity (or any other finding that the authors deem significant) will serve to make the abstract more informative and representative of the data being reported.
We thank the reviewer for this comment. To improve the manuscript, we included the amount of flavonoids, phenolic acids, and saponins in the extract, as well as the compounds that were most prevalent. We also mentioned that the choice of an adequate solvent is important to extract bioactive compounds
- Table 2 lists the total phenolic and total flavonoid content of the analyzed samples, however, its title mentions a “phytochemical screening”. This term appears to be too broad, since only phenolics and flavonoids (a particular type of phenolic) are reported. Please consider renaming this table to more accurately reflect its contents.
We apologize for this error. We changed the title of table 2 to better describe its content. We replaced Phytochemical screening by Total phenolics and flavonoids contents of fruit extracts of C.aegyptiaca.
- In Table 4, please homogenize the number of digits between a value and its error, and be consistent for all data presented on this table.
Indeed, the data of this figure are not homogeneous. We have checked the number of digits between all value and its error according to your recommendations.
- Line 282 states that the samples were “dried for 21 days and then crushed”. Please provide more detailed information about the drying method used (temperature, equipment used, exposure to light/oxygen, etc.). Same comment for the crushing procedure. This is particularly relevant, since the phenolic composition and concentration of vegetable matter may be significantly altered during processing, and could be an important variable for other authors who may study this plant in the future.
We thank the reviewer for this comment. To improve the manuscript, we have expanded this section by adding more informations about the treatment of the samples. We have now included all these details in the manuscript
- In line 287, the crushed fruits are said to have been “extracted successively with 4 L of solvents”. Instead of specifying the absolute volume of solvent used, please provide the mass-to-solvent ratio, in order to make it easier for others to reproduce your extraction method.
We have also indicated the plant material/solvent ratio (w/v) in this section

Reviewer 2 Report
In the presented paper authors have characterized the phenolic compounds and antioxidant activity of extract from Calendula aegyptiaca Fruits. Although the manuscript is well written in general I have some comments and remarks.
The introduction seems to be a summary. Authors should extend this chapter and more information about C. aegyptiaca, if available. Moreover, the last sentences of Introduction should be deleted. The results of the study should not be presented in the Introduction chapter. The aim of the work should also be included in the Introduction.
How the fruits were dried and crushed? These two process have a significant influence on powder properties and consequently on extraction process. What was the degree of the fineness of particles?
The abbreviation should be explained before first time using: for example GAE or QE.
What was the moisture content of dried extracts?
How many points was used to determine each correlation?
Table 5. Description of the Table 5 and under the Table 5. It is written “Pearson's correlation coefficients (R2)” r is a correlation coefficient, R2 is a coefficient of determination.
Author Response
In the presented paper authors have characterized the phenolic compounds and antioxidant activity of extract from Calendula aegyptiaca Fruits. Although the manuscript is well written in general I have some comments and remarks.
- The introduction seems to be a summary. Authors should extend this chapter and more information about aegyptiaca, if available. Moreover, the last sentences of Introduction should be deleted. The results of the study should not be presented in the Introduction chapter. The aim of the work should also be included in the Introduction.
We have taken this remark into account by the last sentence of the Introduction. Furthermore, we did not find in the literature additional information from chemical or biological viewpoints concerning this plant,
- How the fruits were dried and crushed? These two process have a significant influence on powder properties and consequently on extraction process. What was the degree of the fineness of particles?
We thank the reviewer for this comment. To improve the manuscript, we have expanded this section by adding more informations about the treatment of samples. We included the drying and crushing procedures in the paper.
The fruits were crushed until they formed a fine powder. “……placed in the shade in a well-ventilated area with low humidity (22–25%) at a temperature range of 18–25 °C for 21 days and then crushed.”
“The dried fruits are crushed in a grinder from Fritsch Company (reference 14.3000.00) in order to obtain much finer particles (2-3 mm) and then stored in airtight jars away from humidity at room temperature”
- The abbreviation should be explained before first time using: for example GAE or QE.
We apologize for this. As recommended, we checked the description of each abbreviation before its first-time use.
- What was the moisture content of dried extracts?
Indeed, this information is useful. We have included the following information in the paper: “The moisture content of fruits was evaluated to be 19.71%”
“The evaporation process resulted in crude extracts that had no moisture content.
- How many points was used to determine each correlation?
We are sorry, we did not mention the number of replicates. Each test was run at least three times to determine each correlation. We have now added it in the manuscript.
- Table 5. Description of the Table 5 and under the Table 5. It is written “Pearson's correlation coefficients (R2)” r is a correlation coefficient, R2 is a coefficient of determination.
We have taken this remark into account and we have corrected the document. We replaced Pearson's correlation coefficients by Pearson's determination coefficients.

Round 2
Reviewer 2 Report
The authors corrected the manuscript accordingly.